# Bovine Organospecific Microvascular Endothelial Cell Lines as New and Relevant In Vitro Models to Study Viral Infections

**DOI:** 10.3390/ijms21155249

**Published:** 2020-07-24

**Authors:** Anne-Claire Lagrée, Fabienne Fasani, Clotilde Rouxel, Marine Pivet, Marie Pourcelot, Aurore Fablet, Aurore Romey, Grégory Caignard, Damien Vitour, Sandra Blaise-Boisseau, Claudine Kieda, Henri-Jean Boulouis, Nadia Haddad, Catherine Grillon

**Affiliations:** 1UMR BIPAR, Ecole Nationale Vétérinaire d’Alfort, Laboratoire de santé animale d’Alfort, Anses, INRAE, Université Paris-Est, 94700 Maisons-Alfort, France; anne-claire.lagree@vet-alfort.fr (A.-C.L.); clotilde.rouxel@vet-alfort.fr (C.R.); henri-jean.boulouis@vet-alfort.fr (H.-J.B.); nadia.haddad@vet-alfort.fr (N.H.); 2Center for Molecular Biophysics UPR4301 CNRS, 45000 Orléans, France; fabienne.fasani@cnrs-orleans.fr (F.F.); pivetmarine@gmail.com (M.P.); claudine.kieda@cnrs-orleans.fr (C.K.); 3UMR Virologie, INRAE, Ecole Nationale Vétérinaire d’Alfort, Laboratoire de santé animale d’Alfort, Anses, Université Paris-Est, 94700 Maisons-Alfort, France; mariepourcelot@hotmail.com (M.P.); aurore.fablet@vet-alfort.fr (A.F.); aurore.romey@anses.fr (A.R.); gregory.caignard@vet-alfort.fr (G.C.); damien.vitour@vet-alfort.fr (D.V.); sandra.blaise-boisseau@anses.fr (S.B.-B.); 4Laboratory of Molecular Oncology and Innovative Therapies, MIM, 04-141 Warsaw, Poland

**Keywords:** endothelium organospecificity, microvasculature, immortalization, cattle diseases, host-pathogen interactions, viruses

## Abstract

Microvascular endothelial cells constitute potential targets for exogenous microorganisms, in particular for vector-borne pathogens. Their phenotypic and functional variations according to the organs they are coming from provide an explanation of the organ selectivity expressed in vivo by pathogens. In order to make available relevant tools for in vitro studies of infection mechanisms, our aim was to immortalize bovine organospecific endothelial cells but also to assess their permissivity to viral infection. Using transfection with SV40 large T antigen, six bovine microvascular endothelial cell lines from various organs and one macrovascular cell line from an umbilical cord were established. They display their own panel of endothelial progenitor/mature markers, as assessed by flow cytometry and RT-qPCR, as well as the typical angiogenesis capacity. Using both Bluetongue and foot-and-mouth disease viruses, we demonstrate that some cell lines are preferentially infected. In addition, they can be transfected and are able to express viral proteins such as BTV8-NS3. Such microvascular endothelial cell lines bring innovative tools for in vitro studies of infection by viruses or bacteria, allowing for the study of host-pathogen interaction mechanisms with the actual in vivo target cells. They are also suitable for applications linked to microvascularization, such as anti-angiogenic and anti-tumor research, growing fields in veterinary medicine.

## 1. Introduction

Microvascular endothelial cells (ECs) form a continuous monolayer lining the inner face of vascular walls throughout the body and play crucial roles in the regulation of vascular functions (homeostasis, transport of nutrients or hormones, leukocyte adhesion and trafficking across ECs and angiogenesis) [1,2]. The primordial function of the vascular endothelium is to maintain an effective barrier for fluids, proteins, and cells while allowing efficient gas transfer, transport of solutes, and trafficking of inflammatory cells [3]. Alterations of endothelial homeostasis and membrane permeability may lead to pathophysiological outcomes [4]. Indeed, ECs play an important role in wound healing and pathological processes such as inflammation or tumor metastasis [5]. 

Vascular endothelial cells also constitute target cells for microorganisms, in particular for vector-borne pathogens. Pathogens that target ECs can compromise endothelial monolayer integrity and barrier permeability, which is a way for them to cross the vascular barrier, either to reach the blood and disseminate throughout the body or to penetrate inside an organ. Even if infected, ECs are not always the main target cells of pathogens but might represent the site of initial replication after vector-borne inoculation, and before infection of blood cells. Moreover, they could constitute a reservoir for the pathogen during persistent infection by playing the role of niche cells, protecting the pathogen from the immune system [6]. 

Endothelial cells are often targeted by viruses, including members of *Flaviviridae* such as dengue, West Nile or Zika viruses [7], or members of *Reoviridae* such as African horse sickness virus or Bluetongue (BT) virus (BTV) [8], which can result in severe lesions. For instance, Bluetongue is transmitted by hematophagous *Culicoides* midges and is notably characterized in domestic ruminants by vascular injury with hemorrhage and ulceration of the oral cavity and upper gastrointestinal tract, tissue infarction, and widespread edema [9]. Endothelial cells represent the major site of BTV replication, thus explaining the typical lesions that result in excessive bleeding and coagulopathy [8,10]. In the case of Dengue, hemorrhagic fever and shock syndrome are caused by vascular leakage due to impaired endothelial permeability [11]. Additionally, bacteria belonging to the Rickettsiales order, such as *Ehrlichia ruminantium*, a tick-borne bacterium responsible for heartwater in cattle, can also infect ECs [12]. On the contrary, the foot-and-mouth disease (FMD) virus (FMDV), a *Picornaviridae* member, the agent of a highly contagious vesicular disease of cloven-hoofed animals (Artiodactyla order) with a tremendous economic impact at the global level [13], preferentially targets epithelial cells [13,14]. However, it was suggested that microvascular ECs could play immunoregulatory roles in the immune response to FMD vaccines [15]. Thus, more research about FMDV infection of bovine endothelial cells could be of real interest to understand some characteristics of viral pathogenicity. 

Pathogens can express a selectivity towards endothelial cells from a specific organ. In fact, a strong heterogeneity exists between endothelial cells depending on their belonging to the macrovasculature or to the microvasculature but also on their location in the organism. Microvascular ECs isolated from blood vessels of various tissues vary structurally, phenotypically, and functionally according to the organ they are coming from and to their exposure to the microenvironment [16,17]. Their specific gene expression patterns allow them to support functions that are critical for the development and the functions of each particular organ system. Consequently, they tend to show distinct metabolic or phenotypic properties, such as markers expression, angiogenic capabilities, barrier permeability properties but also distinct reactions to pathogen infection [1]. So, the study of pathogen–endothelial cell interactions has to be done with the endothelial cell type that is primarily targeted in vivo.

Despite a constant increase in the number of publications in the fields of veterinary medicine, including cell–pathogen interactions, inflammation, and cancer, valuable biological models are still lacking compared to research tools developed concerning humans and rodents. The use of primary cells requires the use of animals but also displays several disadvantages. First, they stop dividing after a finite and limited number of passages. Second, as batches do not come from the same animal, they differ from one to another and do not provide repeatable and reproducible data. In contrast, immortalized cell lines, established in a controlled and identical manner, can divide infinitely in long-term in vitro culture allowing a large cell production for scientific studies and represent a good alternative to overcome these problems [17,18,19]. 

Only few bovine immortalized cell lines are available up to now, whatever the cell type considered, and cells of human origin, such as human umbilical vein ECs (HUVECs) are even sometimes used as a model of cattle endothelial cells. The most used bovine endothelial cells are originated from the macrovasculature, such as bovine aortic endothelial cells or bovine umbilical cord ECs that allow the growth of the bacterium *E. ruminantium* [20,21,22]. Nevertheless, these cells present strong differences with microvascular ECs that are in close contact with pathogens in vivo and are involved in tumor angiogenesis. For instance, it was shown that the infection of ECs from the aorta or from the organ of interest, the brain, with two isolates of ovine lentiviruses, was very different [23]. Only two bovine immortalized endothelial cell lines from the microvasculature, BCE C/D-1b (ATCC^®^ CRL-2048™) from cornea [24] and EJG (ATCC^®^ CRL-8659™) from the adrenal gland, can be purchased. In addition, only a few immortalized microvascular ECs are described in the literature: bovine brain ECs [25,26], bovine adrenal ECs [27], and bovine luteal ECs [28]. 

To palliate the lack of bovine microvascular endothelial cell lines for veterinary studies, allow the growth of pathogens in adapted cell lines, close to the in vivo conditions, and avoid the regular use of cattle for primary cell needs, there is an urgent need to establish immortalized bovine endothelial cell lines from the microvasculature of various organs representing potential targets for infectious agents. Such organospecific cell lines were previously developed in mice [18], in human (CNRS patent 99-16169) [17] and then in feline [19] models. They allowed the comparison of infection of human versus feline microvascular ECs by *Bartonella henselae*, which can lead to bacillary angiomatosis and/or peliosis in immunocompromised human patients, showing a differential pro-angiogenic effect on human but not feline microvascular ECs. This was consistent with the absence of bacillary angiomatosis in cats and highlighted the fact that the outcome of endothelial cell infection by *B. henselae* is governed by species specificity [19].

In the present paper, our aim is the establishment of bovine endothelial cell lines from various organs as new models to study cattle viral infections. We present the isolation, immortalization, and characterization of six microvascular and one macrovascular bovine endothelial cell lines, that were immortalized in the same way as previous human, murine, and feline ECs, their successful infection by two already mentioned viruses, BTV and FMDV, and their ability to produce viral proteins upon DNA transfection. 

## 2. Results

### 2.1. Establishment of Bovine Endothelial Cell Lines

Six bovine endothelial cell lines from the microvasculature of various organs (brain, intestine, lung, mesenteric lymph node, ovary, and skin) and one cell line from the macrovasculature (from umbilical cord) were established in culture (Table 1). All the stable cell lines display a doubling time between 24 h and 48 h and were then grown for up to 22 passages without any change in cell morphology.

All cell lines displayed endothelial cell characteristic morphologies. Nevertheless, slight morphological differences were observed between the seven cell lines during their growth (Figure 1). Under microscope, bovine mesenteric lymph node microvascular endothelial cells (BMLNMEC) were bigger than the cells from other lines. Bovine brain microvascular endothelial cells (BBrMEC) were very slender, whereas bovine lung microvascular endothelial cells (BLuMEC) and BMLNMEC were stocky cells, and very adhesive between them. All these cell lines, except bovine ovary microvascular endothelial cell (BOvMEC) line, were patented (CNRS/EnvA Patent FR 19 15307) and deposited at the Collection Nationale de Cultures de Microorganismes (Institut Pasteur, Paris, France).

### 2.2. Characterization of Bovine Endothelial Cell Lines

#### 2.2.1. Angiogenesis Assays

Bovine endothelial cell lines were assessed for their ability to form capillary-like structures in vitro in a Matrigel™ assay and compared with the human endothelial cell line (HSkMEC) and the human endothelial progenitor cell line-cord blood (HEPC-CB1) displaying an angiogenic potential. Figure 2 shows that all cell lines were capable of performing angiogenesis (network formation) on Matrigel™, as compared to the reference cell lines HSkMEC and HEPC-CB1. In the case of BBrMEC, bovine intestine microvascular endothelial cells (BIntMEC) and bovine skin microvascular endothelial cells (BSkMEC), the angiogenic process proceeded for less than two hours and then networks progressively deteriorated. Their networks were very similar and looked like HEPC-CB1 network. Concerning BLuMEC, BMLNMEC, BOvMEC, and bovine umbilical cord endothelial cells (BUcEC), the maximum of angiogenic potential was achieved between four and six hours. BLuMEC, and particularly BMLNMEC and BUcEC, formed less pseudovessels but were much thicker, gathering many cells, suggesting a more achieved angiogenesis. Only BOvMEC performed a partial angiogenesis with few vessels and some cells staying isolated.

#### 2.2.2. Identification of RNA Endothelial Cell Markers by RT-qPCR

The seven bovine cell lines were analyzed for the RNA expression of several markers, including endothelial cell markers, progenitor cell markers, and non-endothelial cell markers, using bovine primers (Table 2). All markers were detected for each cell line but with very different levels of expression. To compare the different cell lines and their markers, the calculations were done using each primer pair efficiency and the relative expression of each gene was expressed in comparison to the expression of the three reference genes (*gusb*, *hprt1*, and *ppia*) after a normalization step. 

Concerning mature endothelial cell line markers, CD31, CD143, CD146, VEGFR-2, and von Willebrand factor (vWF) mRNAs were weakly expressed in all bovine cell lines, compared to the mature bovine aortic primary endothelial cells (BAEC), with some differences of expression between the cell lines. In particular, BBrMEC and BMLNMEC displayed the highest expression of VEGFR-2 mRNA and CD146 mRNA, respectively, among all bovine cell lines. In addition, vascular endothelial growth factor A (VEGFA) and vascular endothelial growth factor receptor 1 (VEGFR-1) mRNAs were more expressed on bovine cell lines than on BAEC. Endothelial hyaluronan receptor 1 (LYVE-1) mRNA, a marker of lymphatic endothelial cells, was strongly expressed in the seven bovine cell lines, with a very high relative expression in BOvMEC and BLuMEC, whereas it was very little expressed in BAEC. For the three markers of progenitor cells, the relative expression of CD117 mRNA was high in BBrMEC compared to other cell lines and to BAEC. CD133 mRNA was more expressed in bovine cell lines than in BAEC, particularly in BMLNMEC. Furthermore, chemokine receptor type 4 (CXCR4) mRNA, which can be found both in progenitor and mature endothelial cells, was slightly expressed in the seven cell lines, compared to its high level in BAEC. Finally, CD38 mRNA, a marker of hematopoietic cells, was only slightly expressed, except in BIntMEC.

#### 2.2.3. Identification of Endothelial Cell Markers by Flow Cytometry

To complete the phenotypic study, bovine cell lines were analyzed by flow cytometry for the expression of specific markers at the protein level (Table 3 and Figure 3). As most of the antibodies used were directed against human proteins, controls were performed with BAEC and with human endothelial cell lines, HSkMEC (as mature endothelial cells) and HEPC-CB1 (as progenitor endothelial cells). For endothelial markers, CD143, the angiotensin-converting enzyme (ACE), a typical marker of endothelial cells, was expressed at the protein level in all the bovine cell lines and in the control human cell lines but was weakly displayed in BAEC. CD146, melanoma cell adhesion molecule (MCAM), another typical marker of the endothelium, was present at a high level on BLuMEC, BMLNMEC, BOvMEC, and in all control cell lines including BAEC, and more weakly on BBrMEC. It was totally absent from BIntMEC, BSkMEC, and BUcEC. CD105 (or endoglin) was absent on most assessed bovine cell lines and was weakly present in BAEC, whereas it was highly expressed on human cells. Endothelial nitric oxide synthase (eNOS), a marker of mature endothelial cells, was present in BAEC and HSkMEC and, to a lesser extent, in BLuMEC, BSkMEC, and BUcEC. CD31 was evaluated using antibodies against either the human, the ovine, or the bovine protein and no labeling was observed on any bovine cell line, only a very weak labeling on BAEC with the anti-bovine CD31. As expected with progenitor endothelial cell markers (CD133, CD271, CXCR4), BAEC and HSkMEC displayed the same profile, corresponding to a profile of mature endothelial cells, whereas HEPC-CB1 showed a high expression of all markers. CXCR4, a chemokine receptor, was expressed on all bovine cell lines in a comparative manner to BAEC and HSkMEC. CD133 and CD271 were weakly expressed on BBrMEC, BIntMEC, and BOvMEC, but absent on BLuMEC and BUcEC. BMLNMEC expressed only weakly CD133 while BSkMEC displayed CD271. The hematopoietic markers CD34 and CD45 were not present or only weakly on all cell lines, except CD34 on BLuMEC. In the case of antibodies displaying no or a weak labeling, such as anti-human CD31 and anti-human CD105, it cannot be excluded that the protein was present but that the antibody affinity toward the bovine form of the protein was not sufficient to bind it.

From those results, it was not possible to establish a close correlation between RT-qPCR and flow cytometry data when comparing the expression of markers and their corresponding mRNAs in bovine endothelial cell lines, suggesting post-transcriptional regulations of protein expression.

#### 2.2.4. Detection of VEGFA by ELISA

The secretion of the major endothelial and proangiogenic growth factor, VEGF, was investigated in the culture supernatants of the seven endothelial cell lines as well as of BAEC using three different specific bovine, one murine, and one human VEGFA ELISA detection kits. No VEGFA secretion was detected, whether the cells were grown in normoxia or in hypoxia, a condition well-known to increase VEGFA production.

### 2.3. Permissive Assay to Viral Infection by Two Viruses

Apart from BOvMEC which were not tested, the six other cell lines were infected with BTV at multiplicity of infection (MOI) 0.1. Cells were collected/lysed at 18 h post-infection (hpi) and the expression of the VP5 viral protein was measured by Western blot. As shown in Figure 4, VP5 was expressed in all BTV-infected endothelial cells tested, but BMLNMEC and BUcEC appeared to be much more permissive compared to other endothelial cells.

In another set of experiments, five endothelial cell lines (BIntMEC, BLuMEC, BMLNMEC, BSkMEC, and BUcEC) were inoculated with FMDV strain of O serotype (MOI 10^−3^ and 10^−5^) and monitored for cytopathic effect (CPE) apparition during 48 h. Highly sensitive ZZ-R127 cells were inoculated as positive control. The results are summarized in Figure 5 and Table 4. A CPE was observed as soon as 20 hpi for BMLNMEC cells inoculated with the lowest FMDV dose, similar to what was observed for the ZZ-R127 cells currently used in FMDV diagnostics, for the higher dose. A CPE was observed later, at 48 hpi, for cells derived from umbilical cord and lung. However, the CPE appeared higher for BLuMEC compared to BUcEC. At the opposite, no CPE was observed for endothelial cells derived from the intestine or skin, even after two cell passages. Overall, cells derived from the mesenteric lymph node seemed the most susceptible to FMDV, followed by cells derived from the lung and umbilical cord. However, endothelial cells derived from the skin and intestine do not appear to be susceptible to FMDV infection.

### 2.4. Ability of Endothelial Cell Lines to Express the NS3 Viral Protein upon DNA Transfection

Madin-Darby bovine kidney cell line (MDBK) was used in our laboratory as a model of viral infection since these cells have been notably demonstrated to be permissive to BTV and FMDV infections [29,30]. However, the MDBK cells are very difficult to transfect and do not allow to study the expression, the localization, and the activity of viral proteins inside the cells. To test the DNA transfection process on bovine endothelial cell lines, we carried out fluorescence microscopy in BBrMEC, BLuMEC, BSkMEC, and BUcEC. These cells were transfected with an expression vector encoding BTV8-NS3 fused to the enhanced green fluorescent protein (GFP) and NS3 localization was then analyzed by fluorescence microscopy. At 24 h post-transfection, NS3 was expressed inside all cell lines and localized in specific cytoplasmic structures evocative of the Golgi apparatus and at the plasma membrane (Figure 6), showing thus the successful ability of these four bovine endothelial cell lines to be transfected.

## 3. Discussion

The isolation and maintenance of ECs from specific vascular beds are important to study physiological and pathological processes such as inflammation, wound healing, leukocyte homing and transmigration, tumor development and invasion or metastasis [5], as well as host–pathogen interactions in the cells encountered and/or targeted by pathogens during in vivo infection. Endothelial cells are in close contact with vector-borne pathogens that infect blood cells and they can constitute the niche cells during some of their developmental stages, thus protecting them from the immune system [6]. However, the use of primary cells for such studies needs resorting to animals for isolation and also presents some disadvantages. They cannot be maintained in culture for a long time and they very often lose their phenotype during the culture. By overcoming these problems, immortalized cell lines represent a good alternative to primary cells and constitute crucial tools to develop new in vitro methods for replacing animal use. Indeed, endothelial cell lines allow studies related to endothelial cell infection or other vascular-related diseases, such as tumors. Transfection with the simian virus 40 (SV40), or with human telomerase for human cells, was shown to be efficient to transform cells and establish permanent cell lines that retained characteristics of differentiated cells [31,32]. Such immortalization of human, murine, and feline microvascular and macrovascular ECs was achieved previously [17,18,19,33]. Successful in vitro models of infections of feline and human ECs with *Bartonella henselae*, were then developed and allowed to reproduce at the cell level some of the major differential effects induced by the bacteria in humans versus cats [19].

In the field of veterinary medicine, despite the need of bovine microvascular endothelial cell lines for research purposes, only few microvascular endothelial cell lines from cattle have been established from brain [25,26], adrenal cortex [27], and ovary [28], using either SV40-large T antigen or the human telomerase reverse transcriptase. They are mainly characterized by their morphology, their capacity to undertake angiogenesis, and by the expression of a few specific markers such as CD31, VE-cadherin, von Willebrand factor or receptors for acetylated low-density lipoprotein. Before this study there was no bovine endothelial cell line isolated from the main sites of entry of the viruses into the body such as skin, lung, and intestines and up to now, nothing is known about the susceptibility of the available bovine cell lines to viral infection. 

Here we present the successful isolation, immortalization, and characterization of six endothelial cell lines from the microcirculation of various organs or tissues with potential contact with pathogens (brain, intestine, lung, mesenteric lymph node, ovary, and skin) and one cell line from the macrovasculature (from umbilical cord), obtained from the same bovine fetus. The rapid cell proliferation of the seven selected cell lines in endothelial specific medium and their endothelial cell characteristic morphology were in favor of the successful isolation of the correct cell type. 

The cell lines’ characterization confirmed that they have kept their blood vessel phenotype, with the expression of, at least, several typical EC phenotypic markers such as CD31, CD143, CD146, CXCR4, VEGFA, VEGFR-1, VEGFR-2, and vWF, either at the mRNA and/or at the protein level. The lack of labeling for CD31 and CD105 on both mature endothelial cells, BAEC, and on all the cell lines, may come from the low affinity of the antibody (targeting the human protein) for the bovine form, precluding from concluding about the presence or the absence of the protein. The lack of available bovine cell lines, of anti-bovine specific antibodies, and of valuable bovine ELISA kits emphasizes the paucity of biological tools when researchers work on species other than humans, rats or mice and confirm the importance to develop tools for other animal species, at least for domestic animals, in order to allow advances in veterinary medicine applications. However, CD31, an important marker of ECs, was not detected even when using anti-bovine antibodies, but the corresponding mRNA was present in cell lines, at a level much lower than in BAEC. These results highlight the utility to detect markers at two different levels, especially when antibodies against the species of interest are lacking. LYVE-1, a marker of lymphatic vessels, was detected in all ECs, with a very high expression in BOvMEC and BLuMEC, and to a lesser extent in BBrMEC and BIntMEC, suggesting that these cell lines may be of lymphatic origin or not yet totally differentiated endothelial cells inside the embryo [34,35]. The four markers of endothelial progenitors (CD133, CD271, CXCR4, and CD117) were expressed by the cell lines at various levels. According to these results, the cell lines can be divided into two main groups, one expressing a higher level of CD143 and CD146 both at the mRNA and protein levels, including BBrMEC, BLuMEC, and BMLNMEC, and the second group with a lower expression of both markers including BIntMEC, BSkMEC, and BUcEC. The remaining line BOvMEC is characterized by a high expression of CD143 and CD146 only at the protein level and mainly by the highest expression of LYVE-1. This characterization of bovine cell lines highlights the heterogeneity among cell lines, derived from the same organism but from distinct organs, reflected by the different expression levels of either progenitor endothelial cell or mature endothelial cell markers from one cell line to another.

As compared to BAEC, the seven cell lines expressed to a lower extent the markers that are expressed at a high level in the mature bovine endothelial cells (CD31, CD143, VEGFR-2, vWF, CXCR4, at the mRNA level, and CD146 and eNOS at the protein level). Conversely, they expressed to a higher extent the markers that are expressed at a low level in BAEC (VEGFR-1, LYVE-1, and CD133 at the RNA level), mainly the markers of non-mature endothelial cells. This suggests that the bovine cell lines are not totally differentiated and that they still keep part of their progenitor status. This could be linked to the origin of the cells since the cultures were derived from a calf at six months gestation. At this stage, different endothelial cell populations co-exist, including stem endothelial cells, early or late endothelial progenitor cells, circulating endothelial cells, and mature endothelial cells, displaying a large panel of markers (CNRS patent WO2011086319A1) [33,35,36,37]. 

All the established bovine endothelial cell lines can be cultivated long-term in the laboratory, contrary to primary cells that stop proliferating after a finite number of divisions, and without losing their phenotypic characteristics as primary cells can do. Consequently, they allow the acquisition of repeatable and reproducible experiments. Furthermore, they display their own phenotype, according to the organ they are coming from, allowing to identify their specificity and, thus, providing a relevant in vitro model for organospecific endothelium studies.

In addition to their phenotype, endothelial cell lines can also be characterized by their functional activities. Apart from BOvMEC, whose aptitude to angiogenesis was limited, the six other cell lines demonstrated their ability to form pseudovessels on Matrigel™, confirming their endothelial origin. The fact that cell lines retained their angiogenesis capacity in vitro is capital information, as these cell lines could be used for in vitro studies in cancerology, as a model of hemangiosarcoma, or for anti-angiogenesis applications. The interest of these cell lines is thus not restricted to in vitro infections and host-pathogen interactions, but can be extended to many applications concerning physiological or pathological processes involving endothelial cells.

In order to evaluate the susceptibility of the cell lines to pathogens, their capacity to be infected by two viruses, BTV and FMDV, involved respectively in two ruminant diseases, Bluetongue and foot-and-mouth disease, were compared.

Such cells could constitute good substrates for determining if the significantly higher susceptibility of sheep to BTV as compared to cattle and if the differences in clinical manifestations when cows can become sick (BTV-8 serotype) could be the consequences of variations in endothelial cell susceptibility to BTV infection [10]. Moreover, at the level of one infected host, variations in the susceptibility of vascular endothelial cells between different organs could determine tissue tropism and the location of hemorrhagic lesions [10]. BTV replication was already demonstrated in primary endothelial cell cultures derived from umbilical vein [38], skin capillaries [10] as well as from the pulmonary artery and lung microvasculature of sheep and cattle [9,39,40,41,42,43]. Due to different experimental designs, some results are sometimes conflicting, but the variation in the type of endothelium used could also explain some disparate results [44]. This stresses the importance to develop several endothelial cell lines from the same animal, in order to compare their respective susceptibility to the same pathogen. The comparison of BTV infection between sheep and cattle ECs showed a species-specific answer to BTV infection [40,41,42]. This answer was shown to be based on differences in the production of inflammatory and vasoactive mediators by the two types of cells, which could partly explain why sheep are highly susceptible to BTV-induced microvascular injury, whereas cattle are not [41]. Moreover, Wechsler and McHolland demonstrated that the calf pulmonary artery endothelial cell line (CPAE) was the most susceptible cell line to BTV infection among the 14 ones tested, which is consistent with the in vivo localization of BTV [29]. Nevertheless, infections of immortalized ECs from the microvasculature had never been tested, which could be crucial for this pathogen, given its vector-borne nature. Our work shows that the six microvascular endothelial cell lines tested allow BTV replication, even if the cells derived from mesenteric lymph nodes and umbilical cord seem the most susceptible to infection.

FMDV preferentially targets epithelial cells, in particular in the respiratory tract. In vitro work is rarely based on immortalized bovine cell lines from the primary organs targeted in vivo due to the lack of available cell lines. Bovine cell lines used for FMDV in vitro investigations are indeed epithelial cells derived from kidney (i) MDBK cells used by Kopliku and colleagues [30] or (ii) cells derived from African Buffalo kidney (BK) [45]. The development of these bovine microvascular ECs was the occasion to test their susceptibility to virus infection, even if they are not supposed to be the main cells targeted by FMDV in vivo. A cytopathic effect was observed on three cell lines (BMLNMEC, BLuMEC, and BUcMEC) among the five tested; BIntMEC and BSkMEC not appearing susceptible to FMDV infection, demonstrating thus the differential susceptibility of each bovine cell line. BMLNMEC was the most susceptible cell line, as CPE was visible as soon as 20 hpi. These cells appear to be as susceptible as the ZZ-R127 cells, the cell line isolated from fetal goat tongue epithelial cells currently used in FMDV diagnostics [46]. Thus, BMLNMEC could bring additional information compared to ZZ-R127 goat cells for FMD research in bovine species, even if it is not described for now as the in vivo targeted cell type by FMDV. Interestingly, these results indicate that ECs could contribute to the infection, even if endothelial tropism is not described [47]. To the best of our knowledge, FMDV has never been detected in vascular endothelial cells. Questions remain to be answered about the receptors used by FMDV for infecting these cell lines. 

This work highlights the potentially high susceptibility of BMLNMEC, along with BLuMEC and BUcEC, to pathogen infection or at least to viral infection. This should be further confirmed by testing other pathogens, in particular those targeting endothelial cells at certain infectious stages. In addition, as only some cell lines are permissive to viral infection, it would be interesting to investigate if there is a link between the expression of specific markers of endothelial cell lines and their susceptibility to viral infection.

Furthermore, these cells would be very useful for studying other pathogens that target specifically bovine endothelial cells, like the intracellular bacterium *Ehrlichia ruminantium* [22]. 

In addition to the direct pathogen infection, BBrMEC could bring an interesting solution to study in vitro the blood-brain barrier (BBB) integrity and functionality. Indeed, they could be co-cultivated with other cell types in order to reconstitute an in vitro BBB, which could permit to study the consequences of pathogen infection on the BBB permeability, as was already studied in human cells [48], and more broadly, BBB activity. In addition, our first results about transfection capacity of these cell lines are very promising. This tool is of real interest in animal health research as it can help to localize cellular or pathogen proteins within the cell, or to highlight cellular partners implicated in host-pathogen interactions (e.g., protein co-localization). Finally, besides the study of pathogen-host interaction mechanisms, the seven bovine endothelial cell lines display their own phenotype, thus illustrating the endothelium organospecificity. Therefore, they provide new tools for studying the tropism of pathogens in vitro, allowing the identification of their pathways of entry into the organism, and avoiding the use of whole animal models.

In conclusion, these seven established bovine endothelial cell lines, mainly originating from the microvasculature of various organs, will enrich the available biological tools in cattle research. They constitute a valid and suitable model for in vitro studies as they retain their phenotype after successive passages and can be grown in large quantities. Their organospecificity allows to work with cell lines very close to in situ conditions. They can thus constitute an excellent alternative to animal experimental testing, either as monolayers models or as more complex structures like organoids. Their scope of application will cover not only host-pathogen interaction studies, as we demonstrated here with viral infections, but also pharmacological studies in order to test new chemicals before their in vivo use and cancerology research [36] as they retain their ability to form new vessels. 

## 4. Materials and Methods 

### 4.1. Endothelial Cells Used as Control Cells 

Bovine aortic primary endothelial cells (BAEC), isolated from an Aubrac-cattle aorta, were used as a bovine endothelial cell positive control. This cow was humanely euthanized due to very bad health at the School of Veterinary Medicine of Toulouse, France. BAEC were grown in Dulbecco’s modified Eagle’s medium (DMEM) high glucose supplemented with 10% heat-inactivated fetal bovine serum (FBS), 1% non-essential amino acids solution, 100 U/mL penicillin and 100 µg/mL streptomycin (Gibco-Thermo Fisher Scientific, Illkirch, France), 16 U/mL heparin and 1 µg/mL hydrocortisone (Sigma-Aldrich, Saint Quentin Fallavier, France). The human skin microvascular endothelial cell line (HSkMEC) [49] and the human endothelial progenitor cell line-cord blood (HEPC-CB1) [33] were used as controls for endothelial cells and endothelial progenitor cells, respectively, and were grown in complete improved Minimal Essential Medium (OptiMEM) consisting of OptiMEM supplemented with 2% of FBS, 0.5 µg/mL fungizone, and 40 µg/mL gentamicin (Gibco-Thermo Fisher Scientific). All cells were grown in a humidified atmosphere containing 5% carbon dioxide (CO_2_) at 37 °C and were tested weekly as negative for mycoplasma.

### 4.2. Establishment of Bovine Endothelial Cell Lines

A Montbéliarde female (*Bos taurus*) calf fetus of six months of gestation was humanely removed during a therapeutic cesarean section at the School of Veterinary Medicine in Maisons-Alfort, France. No ethical approval was required in this case, as our experiment joined a cesarean that was performed due to life-threatening condition of the cow. Tissue biopsies were placed in sterile Roswell Park Memorial Institute (RPMI) medium supplemented with 15% FBS, 100 U/mL penicillin, 100 µg/mL streptomycin and 0.8 µg/mL fungizone (Gibco-Thermo Fisher Scientific). Isolation and culture of microvascular and macrovascular endothelial cells from several organs (brain, intestine, lung, mesenteric lymph node, ovary, skin, and umbilical cord) were performed and cell immortalization was undertaken by transfection using the pSV3-neo plasmid containing the SV40 early region of the large T-antigen gene, as already described [17] and CNRS Patent 99-16169. The endothelial cell lines were cultured with complete OptiMEM and were maintained in a humidified atmosphere containing 5% carbon dioxide (CO_2_) at 37 °C. Transfected cells were selected by addition of 40 to 800 µg/mL G418 in the culture medium. Once established, cell line growth was checked at least twice a week using an inverted Leica DMi1 microscope equipped with digital camera Leica MC120 HD. All cell lines were checked weekly for potential mycoplasma contamination and were tested as negative.

### 4.3. Characterization of Bovine Endothelial Cell Lines

#### 4.3.1. Angiogenesis Assays

Matrigel™ matrix (Corning, Amsterdam, The Netherlands) was diluted (1:1; *v*/*v*) in OptiMEM basal medium at 4 °C. Then, 40 µL were distributed in a 96-well plate and allowed to polymerize during 1 h at 37 °C. Cells (1.5 × 10^4^ in 100 µL OptiMEM for each cell line) were seeded on Matrigel™ and cultures were grown at 37 °C in the incubation chamber of the video station equipped with a video microscope (Zeiss Axiovert 200 M, Le Pecq, France) and with an Axiocam camera using the Zeiss Axiovision acquisition software. The direct real-time visualization of the formation of pseudovessels was monitored every 30 min for 24 h. As the kinetics of angiogenesis was not the same for all cell lines, the best time was selected for each of them. The experiments were performed twice, each in triplicate.

#### 4.3.2. Identification of Endothelial Cell Markers by RT-qPCR

After washing with phosphate-buffered saline (PBS), confluent cell monolayers were scraped, lysed, and stored at −80 °C. RNA extraction was performed using NucleoSpin RNA kit (Macherey-Nagel, Düren, Germany) following the manufacturer’s instructions and elution was done in RNase-free water. RNA amounts were determined using the NanoDrop ND-1000 spectrophotometer. Reverse transcription from 1 µg of RNA was performed with random hexamers/Oligo(dT) using Maxima First Strand cDNA synthesis kit for RT-qPCR (Fermentas–Fisher Scientific) according to the manufacturer’s instructions. Specific primers (corresponding to progenitor, endothelial or non-endothelial cell markers) were designed using KiCqStart™ Primers (Sigma-Aldrich). Three reference genes (gusb, hprt1, and *ppia*) were used for normalization. The list of genes analyzed, and the corresponding primer sequences are described in Table 5. PCR reactions were set up in duplicate using LightCycler 480 SYBR Green I Master (Roche Diagnostics, Meylan, France), after a slight modification of the manufacturer’s protocol, allowing the use of 2 µL cDNA per reaction, instead of 5 µL. Reactions were run on a LightCycler 480 (Roche Diagnostics). A dissociation curve was run after each PCR reaction to ensure that only one amplicon was synthetized. The efficiency of each primer pair was calculated using a dilution range of cDNA. The relative quantity of each gene was then calculated by taking into account the efficiency of each primer pair. The geometric mean of reference gene-relative quantities was used to normalize the relative quantities for each gene of interest. Mean and standard deviations of normalized relative quantities were calculated from three independent cell cultures. 

#### 4.3.3. Identification of Endothelial Cell Markers by Flow Cytometry 

Cells were grown until 90% confluence and washed using complete PBS (PBS with 1 mM of CaCl_2_ and 0.5 mM of MgCl_2_) supplemented with 0.5% of bovine serum albumin (BSA) (Sigma-Aldrich) and 0.1% azide (Sigma-Aldrich). Cells were detached using 0.5 mg/mL type I collagenase (Gibco) and washed. Part of the cells was permeabilized with BD Cytofix/Cytoperm (BD Biosciences, Grenoble, France). Cells (5 × 10^5^) were then incubated with either fluorochrome labeled-antibodies for 60 min or with primary antibodies for 60 min, followed by an additional incubation with the fluorochrome-labeled secondary antibodies for 30 min, as described in Table 6. Isotypic controls were done in parallel by incubation with each corresponding immunoglobulin isotype. After washing, stained cells were analyzed by flow cytometry using BD LSR cytometer (BD Biosciences) and results from 10,000 cells were analyzed by Cell Quest Pro software. Results are presented as the difference in fluorescence relative intensity between the cells labeled with the antibody versus the cells labeled with the corresponding isotype, and by the overlay histograms displaying the isotypic control and the antibody labeling, for one representative experiment out of two.

#### 4.3.4. Measurement of VEGFA by ELISA

Cells were grown either in normoxia (18.5% O_2_) or in a hypoxia chamber (1% O_2_, to increase VEGF production) in 6-well plates in complete medium. The complete medium was replaced by medium without FBS 48 h before the collection of supernatants and cultures were grown up to 90% confluence. Supernatants of three independent cultures for each cell line were collected and immediately frozen at −80 °C. VEGFA production was measured using either the human VEGFA kit (R&D, ref DY293B), the murine VEGFA kit (R&D, ref DY493) or three different bovine VEGFA specific ELISA kits (Elisa Genie, ref BOEB0087; Cusabio, ref CSB-EL025833BO; Ray-Bio, ref ELB-VEGFA), according to the manufacturer’s instructions. Each ELISA kit was tested twice with three independent cultures from each cell line. 

### 4.4. Viral Infections of Bovine Endothelial Cells

#### 4.4.1. BTV Infections

Wild-type field strain of serotype 8 (BTV8) is from the National Reference Laboratory collection (ANSES, Maisons-Alfort, France) and was isolated in the Ardennes (France) in 2006 [50]. Then, 5 × 10^4^ cells were dispensed in each well of a 24-well plate, and infected at MOI 0.1 for 2 h at 37 °C in a serum-free OptiMEM. After infection, the infection medium was removed and complete OptiMEM was added. After 18 h, cells were resuspended in lysis buffer (0.5% Nonidet P-40, 20 mM Tris–HCl at pH 8, 150 mM NaCl, and 1 mM ethylenediaminetetraacetic acid (EDTA)) supplemented with Complete Protease Inhibitor Cocktail (Roche Diagnostics) and phosphatase inhibitors (PhosSTOP, Roche Diagnostics). To quantify viral replication, VP5 protein was detected by Western blot analysis using the corresponding monoclonal antibody (Mab 10AE12; Ingenasa). Secondary anti-mouse HRP-conjugated antibody was from GE-Healthcare. Results are presented as one representative experiment out of three.

#### 4.4.2. FMDV Infections

ZZ-R127 (CCLV-RIE 127, FLI) [46], fetal goat tongue epithelial cells, were grown in 45% Iscove’s Modified Dulbecco’s Media (IMDM) with Glutamax^®^ (Invitrogen), 45% DMEM/F-12 (1:1 mixture of Dulbecco’s modified Eagle’s medium (DMEM) and Ham’s F-12) (Invitrogen), supplemented with 10% FBS. Cells were grown at 37 °C in a 2.5% CO_2_ atmosphere and routinely passaged once a week in 75 cm^2^ tissue culture dishes. Bovine endothelial cell lines were grown as already described. FMDV reference strain O Manisa TUR/8/69 (viral titer 6.8×10^6^ plaque forming unit (PFU)/mL) used in this study was provided by the World Vesicular Disease Reference Laboratory (The Pirbright Institute, WWRL, UK).

Cells were seeded into 24-well plates at a density of 5 × 10^4^ cells per well then incubated at 37 °C under 2.5% (ZZ-R127) or 5% (endothelial cells) CO_2_ for 24 h. The day after, monolayers were washed twice with serum-free culture medium before inoculation. Each sample was inoculated in duplicate (100 µL per well, MOI 10^−3^ or 10^−5^). After 1 h of adsorption at 37 °C under 5% CO_2_, inoculums were discarded and replaced by growth medium. Cells were then incubated at 37 °C under 5% CO_2_ and monitored for CPE for 48 h using an inverted LEICA Statif DM IL LED microscope. For each cell line, CPE was estimated qualitatively, by comparison with ZZ-R127 cell line which is very susceptible to FMDV infection [46]. If no CPE was observed after 48 h, monolayers were freeze-thawed at −80 °C, then clarified, followed by a second passage under the same conditions. Cells were considered as non-susceptible to FMDV infection when no CPE was observed after this second passage. Results are presented as one representative experiment out of two.

### 4.5. BTV8-NS3 Transfection Assay 

Next, 5 × 10^4^ cells were dispensed in each well of a 24-well plate (Ibidi μ-plates, Gräfelfing, Germany). One day later, cells were transfected with 0.5 µg/well of pEGFP-C1 expression vector encoding BTV8-NS3 (Lipofectamine 3000, Invitrogen) according to the manufacturer’s instructions. At 24 h post-transfection, the cells were washed three times with PBS, incubated with a 4% paraformaldehyde (PFA) solution (Electron Microscopy Sciences, Hatfield, PA, USA) for 30 min at room temperature and then treated with PBS-glycine (0.1 M) for 5 min at room temperature to quench the cells. The cells were washed three times with PBS and incubated for 30 min at room temperature in a PBS solution containing Hoechst 33258 dye. Preparations were visualized using an Axio Observer Z1 fluorescence inverted microscope (Zeiss). 

## 5. Patents

CNRS/EnvA Patent FR 19 15307: Lignées immortalisées de cellules endothéliales bovines et canines issues de différents organes. Anne-Claire Lagrée, Fabienne Fasani, Clotilde Rouxel, Claudine Kieda, Henri-Jean Boulouis, Nadia Haddad and Catherine Grillon. 

## Figures and Tables

**Figure 1 ijms-21-05249-f001:**
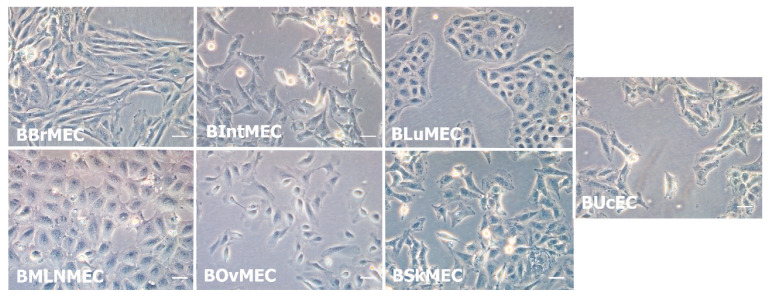
Morphological aspect of the seven established bovine endothelial cell lines. Representative optical microscopy photographs (×20) of endothelial cell lines in culture. Pictures of growing cells were taken using inverted Leica DMi1 microscope equipped with digital camera Leica MC120 HD. Scale bar of 50 μm.

**Figure 2 ijms-21-05249-f002:**
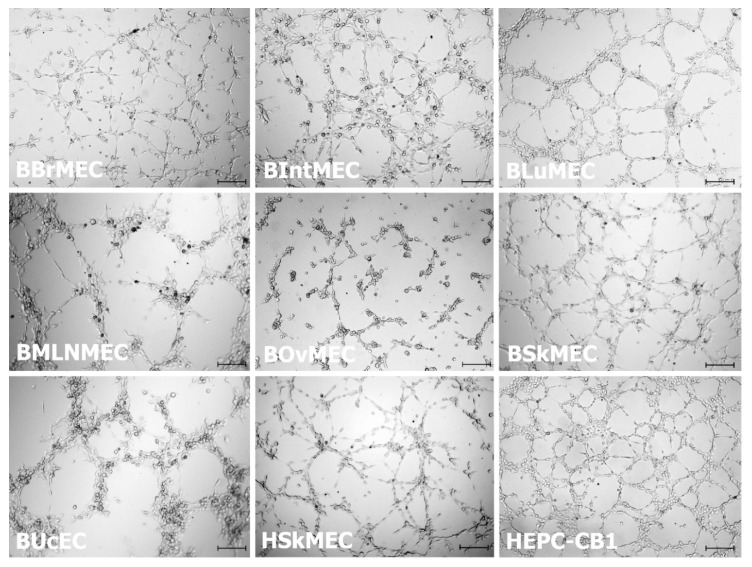
Bovine endothelial cell line angiogenesis assessed by pseudovessel formation on Matrigel™. Cells were seeded on Matrigel™ and angiogenesis was monitored under a video microscope during 24 h. Presented pictures are at 1.5 h (for BBrMEC, BIntMEC, and BSkMEC), at 4.5 h (for BLuMEC, BOvMEC, BUcEC, HSkMEC, and HEPC-CB1) and at 5.5 h (for BMLNMEC) after seeding. Scale bar is 200 μm.

**Figure 3 ijms-21-05249-f003:**
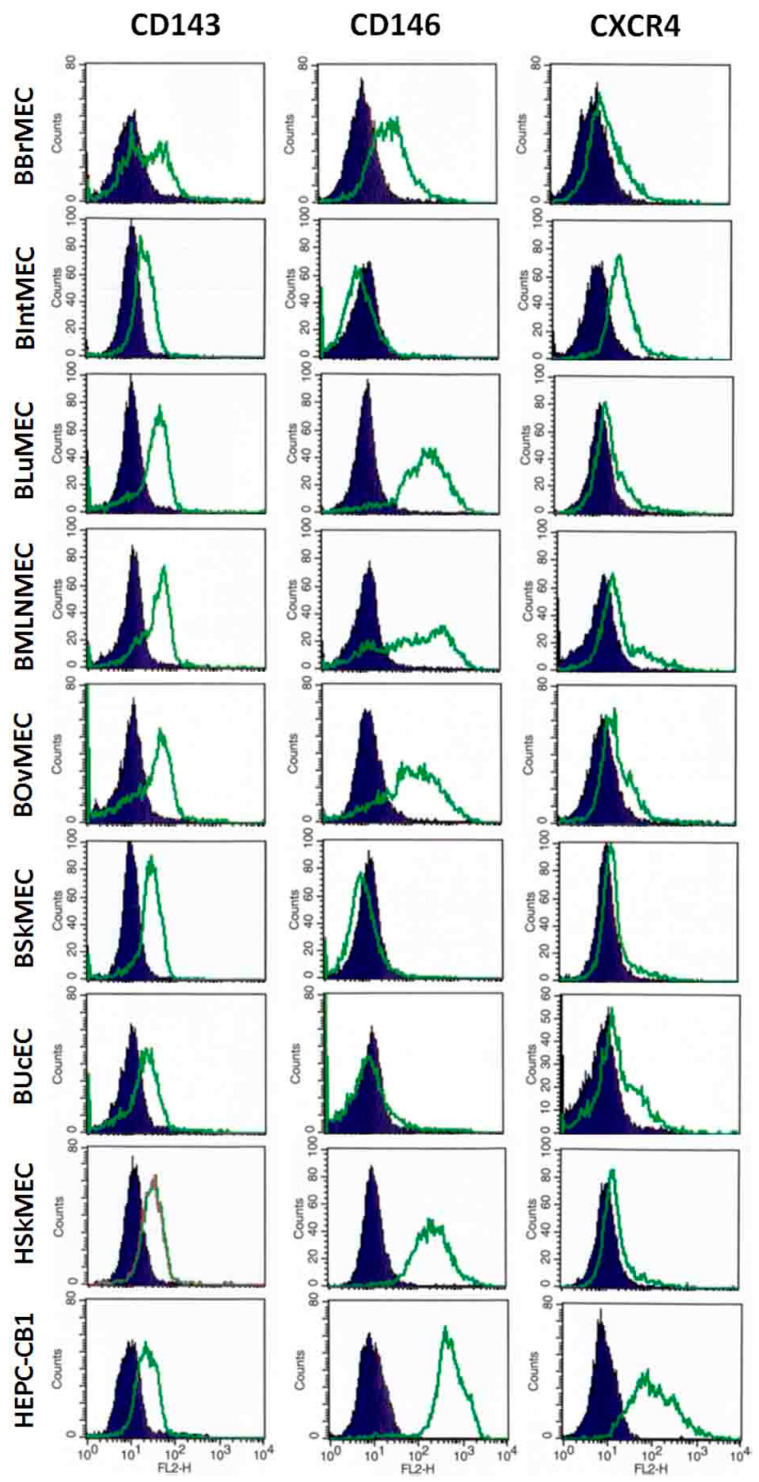
Main endothelial cell markers expressed in various bovine cell lines assessed by flow cytometry. Bovine endothelial cell lines and two human endothelial cell lines, as controls, were labeled with anti-CD143, anti-CD146, and anti-CXCR4 antibodies and then analyzed by flow cytometry. Results are shown as histograms showing the fluorescence intensity with the isotypic control in dark blue and the antibody labeling in green.

**Figure 4 ijms-21-05249-f004:**
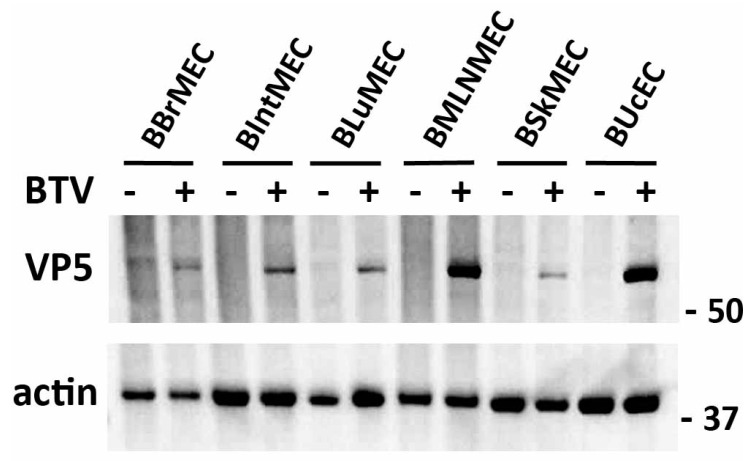
Viral infections of bovine endothelial cell lines by Bluetongue virus (BTV). BBrMEC, BIntMEC, BLuMEC, BMLNMEC, BSkMEC, and BUcEC were infected with BTV (multiplicity of infection (MOI) = 0.1). After 18 h, BTV infection was assessed by anti-VP5 immunoblotting and actin was used as a loading control in Western blot analysis. Originals of Western blot pictures are presented in Appendix A.

**Figure 5 ijms-21-05249-f005:**
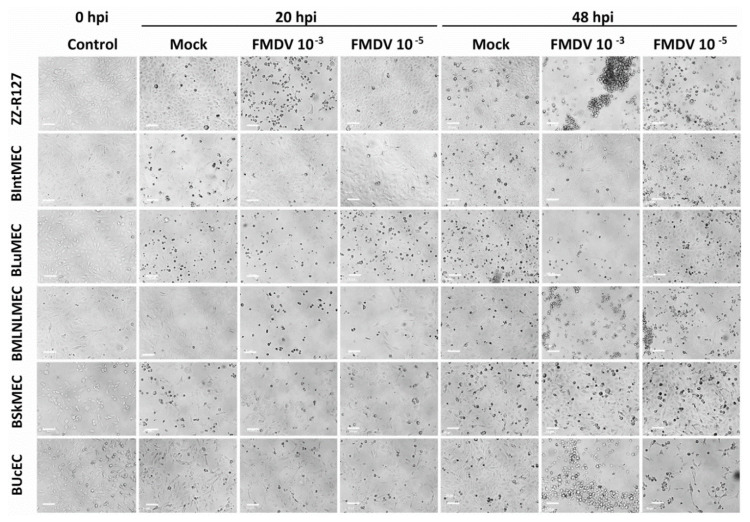
Viral infections of bovine endothelial cell lines by foot-and-mouth disease virus (FMDV). BIntMEC, BLuMEC, BMLNMEC, BSkMEC, and BUcEC were inoculated with FMDV strain of O serotype (2 viral doses: MOI 10^−3^ and 10^−5^) and monitored for cytopathic effect (CPE) during 48 h using an inverted LEICA Statif DM IL LED microscope (×10 magnification). ZZ-R127, fetal goat tongue epithelial cells, were used as positive controls. The pictures presented correspond to 20 and 48 h post-infection (hpi). Scale bars represent 100 μm.

**Figure 6 ijms-21-05249-f006:**
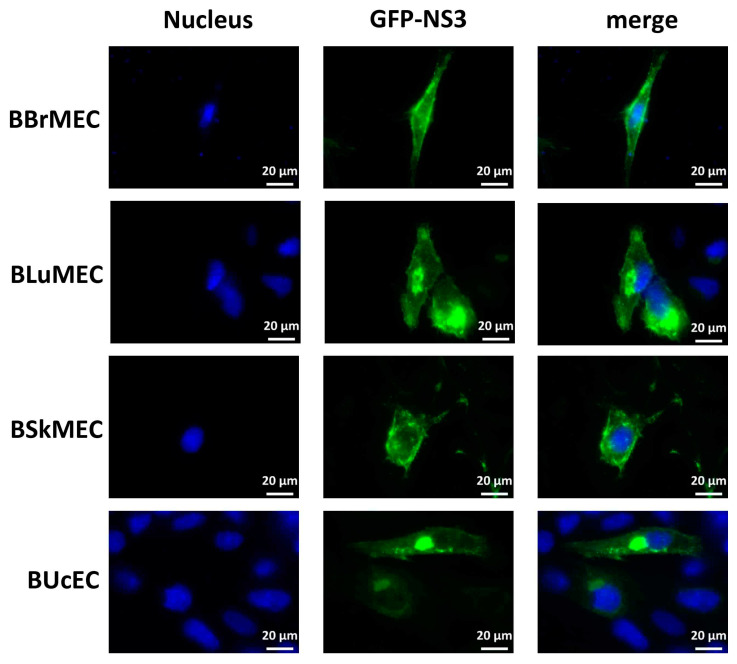
NS3-encoding DNA transfection of bovine endothelial cell lines. BBrMEC, BLuMEC, BSkMEC, and BUcEC were transfected with the expression vector encoding GFP-tagged BTV8-NS3. Intracellular localization of Hoechst-stained nuclei and GFP-tagged NS3 fluorescence was visualized by fluorescence microscopy (×63 magnification) in blue and green, respectively. Scale bars represent 20 μm.

**Table 1 ijms-21-05249-t001:** Established bovine endothelial cell lines.

Organs	Cell Line Names	Meaning	Doubling Time (h)
Brain	BBrMEC	Bovine brain microvascular endothelial cells	48
Intestine	BIntMEC	Bovine intestine microvascular endothelial cells	30
Lung	BLuMEC	Bovine lung microvascular endothelial cells	36
Mesenteric lymph node	BMLNMEC	Bovine mesenteric lymph node microvascular endothelial cells	24
Ovary	BOvMEC	Bovine ovary microvascular endothelial cells	30
Skin	BSkMEC	Bovine skin microvascular endothelial cells	24
Umbilical cord	BUcEC	Bovine umbilical cord endothelial cells	36

**Table 2 ijms-21-05249-t002:** Evaluation of specific RNA marker expression by RT-qPCR in bovine endothelial cell lines and bovine aortic primary endothelial cells (BAEC).

Markers	Normalized Relative Expression	BAEC	BBrMEC	BIntMEC	BLuMEC	BMLNMEC	BOvMEC	BSkMEC	BUcEC
Specific markers for mature endothelial cells									
CD31-PECAM1	mean	2641.39	0.14	0.26	0.04	0.04	0.03	2.41	0.40
	SD	770.92	0.06	0.11	0.01	0.001	0.003	0.06	0.04
CD143-ACE	mean	23.42	2.23	0.74	2.46	0.75	0.05	0.24	0.19
	SD	8.35	1.55	0.32	0.03	0.29	0.01	0.06	0.07
CD146-MCAM	mean	5.55	1.61	0.81	2.24	6.77	0.66	0.07	0.16
	SD	0.44	0.57	0.51	0.20	1.63	0.04	0.01	0.01
VEGFA	mean	0.08	0.79	3.70	1.58	1.05	1.18	0.13	0.68
	SD	0.01	0.27	1.79	0.16	0.35	0.06	0.01	0.04
VEGFR-1-Flt1	mean	0.002	0.86	0.09	0.81	1.17	0.03	1.42	0.47
	SD	0.0009	0.18	0.07	0.09	0.25	0.003	0.02	0.01
VEGFR-2-KDR-CD309	mean	163.16	14.32	0.04	0.11	0.92	0.01	0.01	0.65
	SD	45.71	2.33	0.03	0.04	0.24	0.01	0.003	0.09
vWF	mean	33.09	1.39	2.08	0.13	0.21	0.12	0.06	0.22
	SD	26.91	1.67	1.11	0.03	0.06	0.11	0.01	0.07
Specific markers for lymphatic endothelial cells									
LYVE-1	mean	0.002	36.40	37.31	495.99	2.55	2 461.72	1.92	1.08
	SD	0.001	32.36	23.17	36.94	0.51	547.28	0.17	0.12
Specific markers for progenitor endothelial cells									
CD117-KIT	mean	0.04	19.08	0.0002	0.01	0.001	0.0003	0.0003	0.0001
	SD	0.03	6.32	0.00004	0.01	0.0001	0.0001	0.0001	¤
CD133-PROM1	mean	0.01	1.72	2.13	0.12	4.18	0.41	0.71	0.82
	SD	0.001	0.92	1.29	0.01	1.19	0.02	0.08	0.09
CXCR4-CD184	mean	172.99	1.37	1.97	0.40	1.49	0.02	0.18	2.23
	SD	121.75	0.86	0.22	0.01	0.08	0.004	0.01	0.28
Specific marker for non-endothelial cells (hematopoietic cells)									
CD38	mean	0.008	0.22	4.60	0.05	0.03	0.01	0.66	0.02
	SD	0.006	0.12	4.57	0.01	0.01	0.01	0.05	0.0004

SD: standard deviation; PECAM1: Platelet endothelial cell adhesion molecule; ACE: angiotensin-converting enzyme; MCAM: melanoma cell adhesion molecule; VEGFA: vascular endothelial growth factor A; VEGFR: vascular endothelial growth factor receptor; KDR: Kinase insert domain receptor; LYVE-1: endothelial hyaluronan receptor 1; KIT: proto-oncogene c-Kit; PROM1: prominin-1; CXCR4: Chemokine receptor type 4.

**Table 3 ijms-21-05249-t003:** Detection of specific cell markers in bovine endothelial cell lines by flow cytometry.

Relative Fluorescence Intensity (Arbitrary Units)	BBrMEC	BIntMEC	BLuMEC	BMLNMEC	BOvMEC	BSkMEC	BUcEC	BAEC	HEPC-CB1	HSkMEC
Specific markers for mature endothelial cells										
CD31 (bovine)	<1	<1	<1	<1	<1	<1	<1	4	<1	<1
CD143/ACE	20	9	25	25	30	17	10	1	13	14
CD146/MCAM	44	<1	291	278	172	<1	2	1006	540	265
CD105/endoglin	nt	nt	<1	nt	nt	3	1	5	190	358
eNOS	nt	nt	14	nt	nt	7	7	451	nt	108
Specific markers for progenitor endothelial cells										
CD133	3	4	<1	7	2	2	<1	1	190	1
CD271	5	5	<1	< 1	3	20	<1	5	525	4
CXCR4/CD184	14	29	13	27	12	8	14	9	154	12
Specific markers for non-endothelial cells (hematopoietic cells)										
CD34	nt	nt	10	nt	nt	3	3	3	2	2
CD45	nt	nt	3	nt	nt	2	2	3	2	2

nt: non tested; eNOS: endothelial nitric oxide synthase.

**Table 4 ijms-21-05249-t004:** Cytopathic effect (CPE) observed after FMDV inoculation of bovine endothelial cell lines.

Inoculum	ZZ-R127	BIntMEC	BLuMEC	BMLNMEC	BSkMEC	BUcEC
FMDV 10-3 MOI 20 hpi	CPE +++	Neg	CPE +	CPE +++	Neg	Neg
FMDV 10-5 MOI 20 hpi	Neg	Neg	Neg	CPE+	Neg	Neg
FMDV 10-3 MOI 48 hpi	CPE +++	Doubtful	CPE +++	CPE +++	Neg	CPE +++
FMDV 10-5 MOI 48 hpi	CPE +++	Neg	Neg	CPE+++	Neg	CPE +++

Neg: negative.

**Table 5 ijms-21-05249-t005:** Bovine genes and associated primers used for the cell lines characterization by RT-qPCR.

Primer Names	Associated Genes	Forward Primer Sequence	Reverse Primer Sequence
*gusb*	Beta-glucuronidase	CTGGTTACTACTTCAAGACG	CTGCTTCATAGTTGGTGTTG
*hprt1*	Hypoxanthine-guanine phosphoribosyltransferase	ATCCATTCCTATGACTGTGG	ACTTTTATGTCGCCTGTTG
*ppia*	Peptidylprolyl isomerase A	AAGACTGAGTGGTTGGATG	GTCAGCAATGGTGATCTTC
*cd31*	Platelet endothelial cell adhesion molecule	GAAGACATTATCGGATGCC	TTAATGGCTTCATTGCATGG
*cd143*	Angiotensin-converting enzyme	GAAATGAAACCCACTTTGAC	TCACGAAGTACCTGATATACG
*cd146*	melanoma cell adhesion molecule	CTGGTTTTCTGTCCACAAG	CAGAGTAGTCCCTTTGTCC
*vegfa*	Vascular endothelial growth factor A	GCTGTAATGACGAAAGTCTG	GGAAGCTCATCTCTCCTATG
*vegfr1*	Vascular endothelial growth factor receptor 1	CAACCACAAAATACAGCAAG	GTGACTCTCTCGATAAACAG
*vegfr2*	Vascular endothelial growth factor receptor 2	TGATGAGGAATTTTGTAGGC	ATGGTCTGGTACATTTCTGG
*vwf*	von Willebrand Factor	CAGACACTTCAACAAGACC	TTCCTTGAGTCCTGAAGTC
*lyve-1*	Lymphatic vessel endothelial hyaluronan receptor 1	TACTGCCACAACTCATCTG	GTTGAATAAGGGATCATCGG
*cd117*	proto-oncogene c-Kit	TAGTTCCGTGGACTCTATG	GATGCCAGCTATTCTTCTTC
*cd133*	prominin-1	CGACAGAAGAAAAGTGGTC	CGATGCTTATGAACACACAG
*cxcr4*	chemokine receptor type 4	GATCCGTATATTCACTTCCG	AAGATGATGGAGTAGACAGTG
*cd38*	cyclic ADP ribose hydrolase	TTCATGAGTGCCTTCATTTC	TTTTCGCGTATTCATGAGC

**Table 6 ijms-21-05249-t006:** Antibodies used for cell line characterization by flow cytometry.

Primary Antibody		Corresponding Isotype		Secondary Antibody	
Target	Reference and Provider	Isotype	Reference and Provider	Antibody	Reference and Provider
CD31 (bovine cross reactivity)	MA3100, Invitrogen (Cergy Pontoise, France)	Mouse IgG (serum)	I5381, Sigma-Aldrich (Saint Quentin Fallavier, France)	Goat anti-mouse IgG-PE	Sc-3738, Santa Cruz (Heidelberg, Germany)
CD31(ovine)	MCA1097GA, Bio-Rad (Marnes-la-Coquette, France)	Mouse IgG (serum)	I5381, Sigma-Aldrich (Saint Quentin Fallavier, France)	Goat anti-mouse IgG-PE	Sc-3738, Santa Cruz (Heidelberg, Germany)
		Mouse IgG2a	MCA1210 Bio-Rad (Marnes-la-Coquette, France)		
CD31-APC	130-110-808, Miltenyi Biotec (Paris, France)	Mouse IgG1-APC	555751, BD Biosciences (Le Pont de Claix, France)	*N.A.*	*N.A.*
CD143-PE	FAB929P, R&D Systems (Minneapolis, MN, USA)	Mouse IgG1-PE	IC002P, R&D Systems (Minneapolis, MN, USA)	*N.A.*	*N.A.*
CD146-PE	FAB932P, R&D Systems (Minneapolis, MN, USA)	Mouse IgG1-PE	IC002P, R&D Systems (Minneapolis, MN, USA)	*N.A.*	*N.A.*
CD133-PE	130-090-853, Miltenyi Biotec (Paris, France)	Mouse IgG2b-PE	130-092-215, Miltenyi Biotec (Paris, France)	*N.A.*	*N.A.*
CD271-FITC	130-091-917, Miltenyi Biotec (Paris, France)	Mouse IgG1-FITC	130-092-213, Miltenyi Biotec (Paris, France)	*N.A.*	*N.A.*
CXCR4-PE	FAB170P, R&D Systems (Minneapolis, MN, USA)	Mouse IgG2a-PE	IC003P, R&D Systems (Minneapolis, MN, USA)	*N.A.*	*N.A.*
CD34-FITC	130-081-001, Miltenyi Biotec (Paris, France)	Mouse IgG2a-FITC	130-091-837, Miltenyi Biotec (Paris, France)	*N.A.*	*N.A.*
CD45-FITC	130-080-202, Miltenyi Biotec (Paris, France)	Mouse IgG2a-FITC	130-091-837, Miltenyi Biotec (Paris, France)	*N.A.*	*N.A.*
CD105-PE	FAB10971P, R&D Systems (UK)	Mouse IgG1-PE	IC002P, R&D Systems (UK)	*N.A.*	*N.A.*
eNOS	160880, Cayman chemical (Hamburg, Germany)	Rabbit IgG (serum)	I5006, Sigma-Aldrich (Saint Quentin Fallavier, France)	Goat anti-rabbit IgG-PE	Sc-3739, Santa Cruz (Heidelberg, Germany)

All antibodies target the human proteins except when this is mentioned. N.A.: not applicable.

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
