# Peer review of "Bovine Organospecific Microvascular Endothelial Cell Lines as New and Relevant In Vitro Models to Study Viral Infections"

_ijms, 2020, doi:10.3390/ijms21155249_

Round 1

Reviewer 1 Report

I don’t have any major concerns about current manuscript.

Here are some minor comments:

  • Please improve the quality of the Figures 1 and 2. The lighting of the pictures is quite uneven.
  • Do you have any results about effects of viral replication in the newly generated cell lines (e.g. induction of apoptosis, cytokine release etc.)?
  • My suggestion is that you delete the part of Covid-19 in the discussion. I understand perfectly well your intention. But from my point of view this paragraph doesn’t really fit into the text.

Author Response

The authors would like to thank the reviewer for his interesting comments that will help us to improve our manuscript.

Here are some minor comments:

  1. Please improve the quality of the Figures 1 and 2. The lighting of the pictures is quite uneven.

Figures 1 and 2 have been modified according to your comments, to improve their quality. In Figure 2, the background cannot be totally homogenous as the cells were on the Matrigel matrix and this induced heterogeneity in lighting. The new figures have been introduced in the manuscript.

  1. Do you have any results about effects of viral replication in the newly generated cell lines (e.g. induction of apoptosis, cytokine release etc.)?

In this paper, we only checked the susceptibility of the different endothelial cell lines to viral infection with the two viruses tested. We have not gone further in this work yet. Studying viral infection and its consequences at the cellular level, as you suggested, will be very interesting and should be the next step of this project, including its effects on the production of type I interferon, other cytokines and apoptosis for both viruses.

  1. My suggestion is that you delete the part of Covid-19 in the discussion. I understand perfectly well your intention. But from my point of view this paragraph doesn’t really fit into the text.

According to your suggestion, this part was deleted from the manuscript.

Reviewer 2 Report

I found this manuscript quite interesting. The established cell lines have potential to improve our knowledge of virology both in veterinary medicine and human health sciences. 

Could you please authors comment on:

  1. The kinetics of main endothelial cell surface expression in the cell lines assessed by flow. Any changes during the passages? 
  2. Is there any correlation between the levels of main endothelial cell surface marker expression and susceptibility to the viral infectious? 
  3. Any signs of the involvement of interferon type I signalling in the observed viral infection?  

Author Response

I found this manuscript quite interesting. The established cell lines have potential to improve our knowledge of virology both in veterinary medicine and human health sciences.

The authors would like to thank the reviewer for his interest in our work and for his relevant comments that will help us to improve our manuscript.

Could you please authors comment on:

  1. The kinetics of main endothelial cell surface expression in the cell lines assessed by flow. Any changes during the passages?

The experiments were performed at passages 7-12 and 24-28, according to the cell lines, without any change in the expression of markers when comparing the different cell lines to each other (as the cytometer was changed in the meantime we were not able to directly compare the fluorescence intensities).

  1. Is there any correlation between the levels of main endothelial cell surface marker expression and susceptibility to the viral infectious?

This is an important point that we did not address in the manuscript. Following your suggestion, we went back to cell characterizations. According to our results and the markers we studied, it is difficult to find some markers that allow making a distinction between the respective permissivity of endothelial cells to viral infection. Answering your question would likely require a more extensive study which is beyong the scope of the present paper. However, such a study would be very interesting if we identify some markers that could be potential candidates.

This is why we added the following sentence in the text (page 14, line 423): “In addition, as only some cell lines are permissive to viral infection, it would be interesting to investigate if there is a link between the expression of specific markers of endothelial cell lines and their susceptibility to viral infection.”

  1. Any signs of the involvement of interferon type I signalling in the observed viral infection?

In this paper, we only checked the susceptibility of the different endothelial cell lines to viral infection with the two viruses tested. We have not gone further in this work yet. Studying viral infection and its consequences at the cellular level, as you suggested, will be very interesting and should be the next step of this project, including its effects on the production of type I interferon, other cytokines and apoptosis for both viruses.